# Cryogenic Systems for Astronomical Research in the Special Astrophysical Observatory of the Russian Academy of Sciences

**Yurii Balega** [1], **Oleg Bolshakov** [2], **Aleksandr Chernikov** [2,3], **Valerian Edelman** [4], **Aleksandr Eliseev** [2], **Eduard Emelyanov** [1], **Aleksandra Gunbina** [2], **Artem Krasilnikov** [1,2], **Ilya Lesnov** [2], **Mariya Mansfeld** [1,2], **Sergey Markelov** [1], **Mariya Markina** [4], **Guram Mitiani** [1], **Evgenii Pevzner** [2], **Nickolay Tyatushkin** [2], **Gennady Valyavin** [1], **Anton Vdovin** [1,2] and **Vyacheslav Vdovin** [1,2,*]

1 Special Astrophysical Observatory Russian Academy of Sciences, Nizhnij Arkhyz 369167, Russia; balega@sao.ru (Y.B.); eddy@sao.ru (E.E.); gvalyavin@sao.ru (G.V.); vdovinav@ipfran.ru (A.V.)
2 A.V. Gaponov-Grekhov Institute of Applied Physics Russian Academy of Sciences, Nizhniy Novgorod 603950, Russia; chern@nf.jinr.ru (A.C.); gunbina@ipfran.ru (A.G.); lesnov@ipfran.ru (I.L.); ttshkn3@bk.ru (N.T.)
3 Joint Institute for Nuclear Research, Dubna 141980, Russia
4 P.L. Kapitza Institute for Physical Problems Russian Academy of Sciences, Moscow 119334, Russia; edelman@kapitza.ras.ru (V.E.); mamarkina@edu.hse.ru (M.M.)
* Correspondence: vdovin@ipfran.ru

**Abstract:** This article presents the main results and new plans for the development of receivers which are cooled cryogenically to deep cryogenic temperatures and used in optical and radio astronomy research at the Special Astrophysical Observatory of the Russian Academy of Sciences (SAO RAS) on both the Big Telescope Alt-Azimuthal optical telescope (BTA) and the Radio Astronomical Telescope Academy of Sciences (RATAN-600) radio telescope, 600 m in diameter. These two instruments almost completely cover the frequency range from long radio waves to the IR and optical bands (0.25–8 mm on RATAN and 10–0.3 μm, on BTA) with a certain gap in the terahertz part (8–0.01 mm) of the spectrum. Today, this range is of the greatest interest for astronomers. In particular, the ALMA (Atacama Large Millimeter Array) observatory and the worldwide network of modern telescopes called the EVH (Event Horizon Telescope) operate in this range. New developments at SAO RAS are aimed at mastering this part of the spectrum. Cryogenic systems of receivers in these ranges are a key element of the system and differ markedly from the cooling systems of optical and radio receivers that ensure cooling of the receivers to sub-Kelvin temperatures.

**Keywords:** optical and radio astronomy; cryogenic receivers; cryogenic systems; radio waves; IR and optical ranges; terahertz range; superconductivity; BTA (Big Telescope Alt-Azimuthal); RATAN-600 (Radio Astronomical Telescope Academy of Sciences—600 m in diameter)

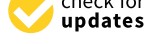



## 1. Introduction

The modern progress of astronomy, with its extremely weak signals and extremely remote objects of studies, imposes most ambitious requirements on receiving equipment. These requirements are several orders of magnitude higher than those for the ground-based receivers used in communications, radar, etc., and actually determine the level of development of modern instrumentation. The main characteristic of astronomical receivers is their sensitivity. Achieving extremely high sensitivity leads almost inevitably to the need for deep cryogenic cooling of the input elements of astronomical receivers.

Initially, astronomy developed in the optical range, and since the second quarter of the 20th century, in the radio range as well. The last century of the development of astronomy was characterized by the expansion of the spectrum and the filling of the space between the optical and radio ranges [1,2], as well as an advance towards shorter wavelengths than optics [3–5]. The critical limitation in this range was not only the technology of the

equipment with its fundamental quantum limitations, but also the atmosphere, which affects the propagation of electromagnetic waves [6].

As a result of the development of the equipment, ground-based astronomy went partially into space [7–9], where there was no limitation due to the atmospheric influence. But astronomy is still impossible without ground-based research, not to mention the huge price and short life spans of space observatories. Therefore, there is a boom in the construction of new observatories around the world.

This work presents the development of cryogenic systems for radio astronomy research at telescopes of the SAO RAS. The following two subsections of the introduction define SAO's place among modern observatories, describe the main types of cryosystems for astronomy and some of SAO's instruments using cryogenics, and present the atmospheric conditions of the site for observations. The next section presents the original developments of cryosystems for telescopes of SAO RAS. In Section 3, we describe the latest developments of the authors related to the field of the sub-THz astronomy technology, which is being actively developed in the world recently. The results of creating superconducting radio detectors cooled to sub-K temperatures for use as elements of the BTA optical telescope are presented.

### 1.1. The Special Astrophysical Observatory of the Russian Academy of Sciences among Astronomical Observatories: Instrumentation and Astroclimate

SAO RAS [10] has two telescopes in its arsenal: the 600 m RATAN-600 radio telescope, which covers a wide range of radio waves up to millimeter waves; and the BTA (Big Alt-Azimuth) telescope, a 6 m optical reflector that successfully operates in the visible and infrared parts of the spectrum. In fact, a part of the spectrum called the terahertz gap remains unfilled. To fill this gap, specialized antennas, good astroclimate, and highly sensitive receivers are needed. Atmospheric absorption of the THz waves is so great that there are only limited opportunities for ground-based astronomy to work in its low-frequency part (0.1–1.0 THz), which is called the sub-THz range. At the lower boundary of the band (100 GHz), the atmosphere gives not very significant absorption almost everywhere, and observatories are located even on the seashore [11]. The upper edge of the sub-THz range is open to ground observations only in unique places on Earth, such as the Atacama Desert [1,2], Pamir [12], Tibet [13], Mauna Kea, Hawaii [14], and the Antarctic Plateau [15]. In the central part of the sub-THz range, the atmospheric transparency windows are 1.3 mm (~230 GHz) (the operating range of the current Event Horizon Telescope (EHT) [16]) and especially 0.8 mm (~375 GHz) (the EHT perspective window), which is highly critical for choosing the location of the observatory. Usually, these are mountains from 3 km above the sea level and above in dry climate and remote from large water bodies.

EHT is a global network of radio telescopes operating together on the Very Long Baseline Radio Interferometer (VLBI) principle. VLBI allows you to combine data from several remote receivers. Using VLBI, multiple independent radio antennas separated by hundreds or thousands of kilometers can act as a phased array, a virtual telescope that can be electronically aimed with an effective aperture equal to the diameter of the planet, greatly improving its angular resolution. Moreover, the wider and more complete the coverage of the UV plane of the network on the Earth's surface is, the higher is the quality of the final result. Today, the plans to expand the EHT VLBI base are aimed at filling the blank spot on the map of the Earth, specifically, the northeastern part of Eurasia. Our project will follow this trend. And the fact that this is a pilot project will determine the construction of a full-scale sub-THz telescope in this part of the planet.

In 2019 [16], one of the most impressive achievements in millimeter-scale astronomy was obtained: for the first time, an image of the shadow of a supermassive black hole in the M87 galaxy was obtained. Publications [17] provide an overview of the plans to expand EHT and highlight the North Caucasian region as promising and requiring a new telescope.

SAO RAS does not have an exceptional astroclimate [18,19], and the altitude above sea level is not very significant: BTA is at an altitude of 2100 m, and RATAN-600 is at an

altitude of 970 m. The astroclimate is quite satisfactory for observations in the entire radio range of RATAN-600 observations down to millimeter waves. The location of BTA has a satisfactory astroclimate in the optical and near-infrared ranges. There is a significant level of cloudy days. Sub-THz waves have a noticeable feature, and even the upper site of SAO RAS, where BTA is located, is far from the best sub-THz observatory site in terms of atmosphere transparency. Moreover, even in the Caucasus, there are more promising sites in Dagestan, but not only the tools, but also the infrastructure are not there yet.

At the same time, the astroclimate studies [18,19] make it possible to work reliably in the two lower sub-THz transparency windows (3 mm and 2 mm) without any problem. There are reliable and rather long periods in winter when there is good atmospheric quality of around 0.1 Np absorption at the zenith in the 1.3 mm transparency window. Sub-THz transparency windows of 0.8 and shorter are practically closed for observations from the BTA site.

More than 50 years of development of astronomy and the equipment for it in SAO were accompanied by the development of cryogenic systems of various types and for different temperatures, providing cryogenic cooling and high sensitivity of receivers. The cryosystems at SAO operate in the temperature range of liquid nitrogen (~70 K), hydrogen (~20 K), and helium (~4 K). An extensive list of similar developments of cryosystems is presented in publications of well-known observatories, for example ALMA or Sardinia [20]. A more detailed overview of the achievements and comparison with our developments will be presented in [21].

The construction of a sub-THz telescope, starting with the selection of a site with low atmospheric absorption, is an expensive and time-consuming undertaking. At the same time, since the beginning of the 1990s, attempts have been made to use sub-THz receivers on the BTA optical telescope. The size of the mirror for a radio telescope is rather small, and the astroclimate is not the best for sub-THz waves, but the quality of the surface exceeds the requirements of THz waves by two orders of magnitude. An important feature of bolometric receivers in the sub-terahertz range is the need for cryogenics at the sub-Kelvin level. This article is concerned with the analysis of already completed and ongoing developments of various cryogenic systems for the RATAN-600 and BTA telescopes of the SAO RAS, as well as on their basis for other telescopes. These systems for heterodyne radio receivers and optical photodetectors had temperatures no lower than the temperature of liquid helium. The article also presents a new project being developed for the Special Astrophysical Observatory of the Russian Academy of Sciences, specifically a radio receiver as part of the instruments of an optical telescope. A matrix bolometric receiver was chosen, rather than a heterodyne. Therefore, a new challenge for the project, a cryogenics system at the sub-K level of temperature, is presented here.

### 1.2. Main Types of Cooling Systems for Astronomical Receivers

The both main types of cooling systems are used to cool astronomical radiation receivers in SAO RAS: accumulators (Dewar vessels) and refrigerators (containing a working refrigerator). However, astronomers have long preferred refrigerators that do not require monitoring and periodic refueling with a cryoagent. In addition to the principle of cooling, refrigeration systems differ in temperature levels. In the optical range, noncryogenic cooling systems (above 100 K) are widely used. Typically, this type of cooling uses thermoelectric coolers to reduce the dark current of the CCD receiver arrays. Nitrogen level cooling (~80 K) is also quite popular for cooling optical and infrared photodetectors (e.g., [22–24]). This is provided both by Dewars filled with liquid nitrogen and nitrogen level coolers, which are relatively cheap and commercially available [24], and usually built on the Stirling cycle.

Since the 1970s, refrigerators with a closed cycle of the hydrogen temperature level (~20 K), built on the basis of the Gifford–McMahon thermodynamic cycle, have been widely used in astronomy [25,26]. In wide practice in astronomy, there is no information on the use of a liquid hydrogen cryostat filler. However, such cryostats were developed for other purposes, and liquid hydrogen is much cheaper than helium used for cooling systems.

Moreover, according to experts, liquid hydrogen is a much calmer and more convenient gas to work with [27], but the prejudice of its being explosive does not allow them to be used in astronomy still. Refrigerators of the hydrogen temperature level, but operating with helium, have so far reached the almost-helium temperatures (5–6 K) and are actively used in astronomy for cooling nonsuperconducting microwave receivers based on Schottky barrier diode (SBD) detectors and high electron moving transistor (HEMT) amplifiers. Classical helium systems (operating at a temperature level of about 4 K) with cooling both in the filler configuration and in the refrigerator have found wide application in ground-based astronomy. This temperature level makes it possible to ensure reliably the transition to the superconducting state of the main structural materials (Nb, Pb, NbN, etc.), on the basis of which various types of modern superconducting heterodyne receivers are developed. Along with the Gifford–McMahon cycle, pulsed tubular coolers that do not contain moving parts in the cooling head have been used actively in recent decades. This has radically reduced the level of vibration being the main disadvantage of refrigeration cooling systems as compared to cryoaccumulators, i.e., Dewars.

Helium systems, both filled by pumping and cooled, can reach temperatures below 4 K (about 1.6–2.0 K). However, in order to overcome the milestone at 1 K, other approaches and other technologies are required. Today, sub-Kelvin (<1 K) temperature levels in astronomy are provided by cryosorption systems (0.3 K) and dilution cryostats (0.01 K). These cycles are implemented based on precooling systems up to 4 K using a refrigerator or liquid helium. Sub-K temperatures are necessary to cool the most sensitive receiving systems, e.g., direct bolometric detectors and their arrays.

Almost all of these systems have been implemented over the past 50 years of development of cooled receivers for the BTA and RATAN-600 telescopes in various configurations and are presented in the following sections. Some of them are currently under development and will be used for the BTA observations in 2025. In this work, we focused on examples of the development of cryogenic receivers for telescopes of the SAO RAS. A more complete list of our results, including equipment for other devices, as well as a more detailed overview of the world practice of creating such systems, is presented in our second publication [21], which will be published after this one.

## 2. Materials and Methods: Development of Cryogenic Systems for Optical and Radio Astronomy at SAO RAS

### 2.1. Cooling Systems for CCD Arrays

CCD arrays (charge-coupled devices) are currently the main working tool in optical astronomy, including the visible, IR, and UV ranges. One of the most effective ways to improve noise characteristics and eliminate the dark current, and hence increase the sensitivity of the receiving device, is to cool it down. Below is a line of various cooling systems operated at different temperature levels and based on different principles of cooling CCD arrays for BTA and other instruments, created in cooperation with SAO RAS and IAP RAS for more than 30 years. Some of the first results are presented in review [22].

#### 2.1.1. Nitrogen-Cooled Optical Cryostat

The main advantages of the presented product are the vibration-free system and the liquid-free cryostat. See Figure 1 for photos.

The main characteristics of the product are as follows:

- T = 80 K;
- Vacuum level is $10^{-4}$ mbar (at least 3 months);
- Overall dimensions: diameter 224 × 464 mm;
- Vacuum camera dimensions: 160 (diameter) × 80 mm (height);
- Optical window: window with a diameter of 109 mm, window material is quartz.

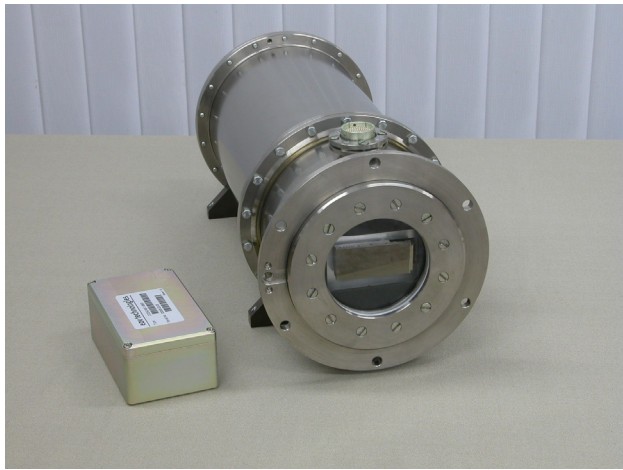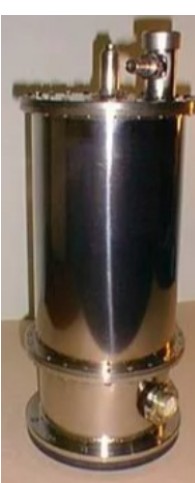

**Figure 1.** Photos of an optical vacuum chamber with nitrogen cooling. Operating temperature T = 80 K.

2.1.2. Optical Vacuum Chambers with Cooling by the CryoTiger Cryogenic System

The created optical vacuum chambers are intended for cooling the receivers used on the BTA telescope [10]. Two chambers were developed, which are cooled by one and two microcryogenic systems (the photos are shown in Figure 2; the main characteristics and advantages are given in Table 1). The main feature of the presented devices is the maintenance of a stable operating temperature of 80 K with extra-large optical windows with a diameter of 220 mm and 326 mm. The presence of optical windows of this diameter makes it possible to accommodate a sufficiently large number of photodetectors (a very large matrix).

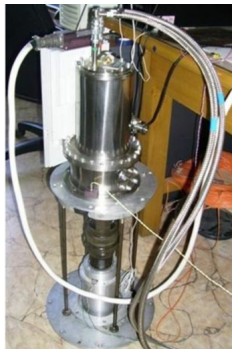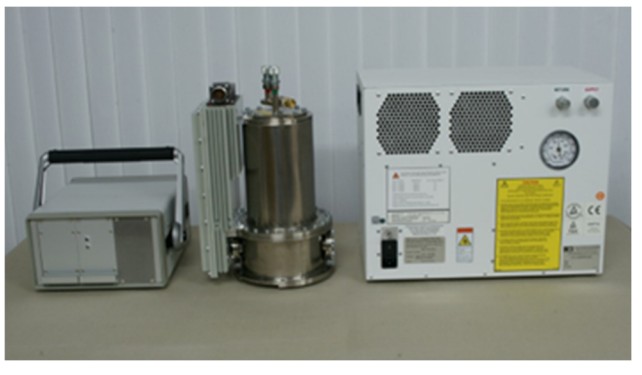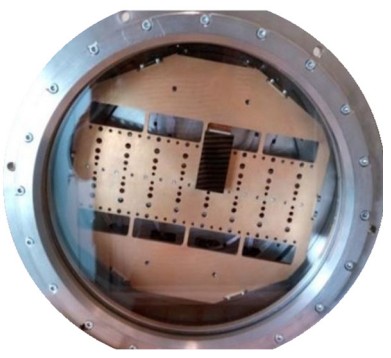

(**a**)                                      (**b**)

**Figure 2.** Photos of optical vacuum chambers with CryoTiger microcryogenic cooling system: (**a**) cooling by the CryoTiger; (**b**) cooled plate cooled by two CryoTigers.

The uniqueness of the design with two CryoTiger [28] units is the fact that two identical cryogenic machines work in parallel for one cooling unit. Usually, this problem does not have an accurate solution due to the incomplete identity of the refrigerators and their operating conditions. The authors managed to solve this problem and almost double the cooling capacity of the system by selecting identical refrigerators and controlling their modes using adjustable thermal switches on cryogenic interfaces. An additional benefit is the increased reliability. In case of failure of one of the machines, the second one will provide cooling at a slightly higher level, which will further reduce the dark current of the photodetector, compared to the uncooled version.

**Table 1.** Main characteristics and advantages of optical vacuum chambers with the CryoTiger microcryogenic cooling system.

|  | **Cooling with One CryoTiger** | **Cooling with Two CryoTiger Units** |
|---|---|---|
| Main Features | - Operating temperature: T = 80 K;<br>- Vacuum level is $10^{-4}$ mbar;<br>- Overall dimensions:<br>- Diameter—300 × 430 mm;<br>- Dimensions of the cavity:<br>- Diameter is 258 × 100 mm;<br>- Optical window:<br><br>Diameter—220 mm;<br>Material—quartz. | - Operating temperature: T = 80 K;<br>- Vacuum level is $10^{-4}$ mbar;<br>- Overall dimensions:<br>- Diameter—388 × 533 mm;<br>- Dimensions of the cavity:<br>- Diameter is 344 × 140 mm;<br>- Optical window:<br><br>Diameter—326 mm;<br>Material—quartz. |
| Advantages | - Large optical window;<br>- The presence of a detector suspension unit made of hard-to-process material<br><br>(superinvar);<br><br>- The possibility of placing and cooling a large number of photodetector devices (CCD arrays). | - Extra-large optical window with a diameter of 326 mm;<br>- The possibility of placing and cooling a large number of photodetector devices (CCD arrays). |

### 2.1.3. Chambers with Thermoelectric Modules for Cooling Large-Format CCD Arrays

Cooling without the use of liquid or compressed gases seems very attractive. This opportunity is provided by thermoelectric, thermomagnetic and laser cooling systems. In practice, there is only experience in using Peltier elements. And there are two radical limitations here. The temperature was supposed to be no lower than 150 K. Moreover, it is achieved by cascading a fairly significant number of Peltier elements (up to 7–8) and extremely low efficiency, the need to remove a significant heat flow from the hot junction of the thermoelectric battery. The photos of the created chamber cooled by Peltier batteries are presented in Figure 3.

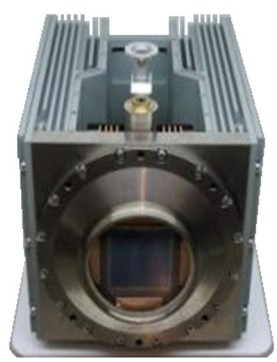
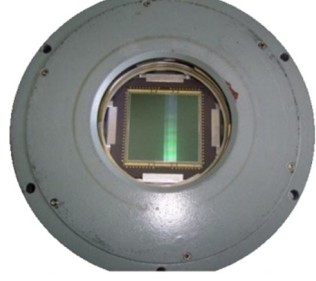
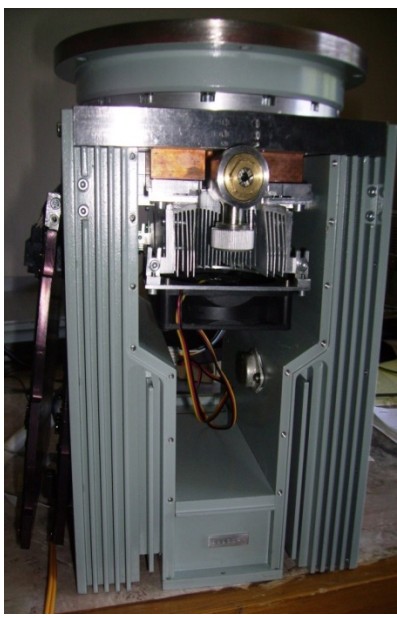

**Figure 3.** Photos of the created chamber with cooling by thermoelectric modules.

The main characteristics of the product are as follows:

- T = 150 K;
- Vacuum level is $10^{-4}$ mbar;
- Dimensions of the vacuum cavity: $174 \times 176 \times 89$ mm;
- Optical window:
- Diameter—90 mm;
- Material—quartz.

2.1.4. Nitrogen Optical Cryostat for Cooling the IR Spectrophotometer

To expand the range of BTA observations into the near-IR range (wavelengths of 0.8–2.5 microns), a unique nitrogen optical cryostat was developed for cooling the IR spectrophotometer. In this article, we will only briefly present the description and main characteristics of the cooling system. The receiving device is placed in a cylindrical Dewar, Figure 4a.

A part of the cryostat is occupied by a no-spill container with liquid nitrogen intended for operation at the primary focus of BTA. Similar designs were implemented for other BTA tasks. The optical circuit contains only two areas that require deep cooling, specifically, the light receiver and the output iris with diffraction gratings. The high stability of the maintained temperature is needed only for the light receiver. The nitrogen tank is located in the upper part of the cryostat and is a cylinder with an asymmetric recess. The shell and top covers of the nitrogen tank are made of stainless steel. The bottom ring of the container with heat-removing contact pads is made of copper. The nitrogen tank is fixed inside the cryostat body by welding an inner filling tube to it (which in turn is welded to an external nitrogen tube, which is also welded to the outer flange of the cryostat). In addition, the lower part of the shell of the nitrogen tank is stretched with Kevlar threads to ensure the desired rigidity of the structure. The cooling of the main elements of the optical circuit is carried out by means of flexible copper cooling wires attached to the copper rim of the nitrogen tank. To ensure the operation of the cryostat at any angle of inclination, the tank covers are connected inside it by copper cooling wires. All optical and electromechanical equipment is placed on a welded stainless steel frame attached to the lower flange of the device through heat-insulating gaskets (Figure 4b).

Main characteristics of the cryogenic system:

- Minimal temperature is T = 80 K;
- Filler nitrogen cryostat with a nitrogen tank of complex shape;
- All elements of the optical scheme of the IR spectrophotometer are cooled to different temperatures inside the cryostat, including moving parts;
- System dimensions: diameter 572 mm, height 660 mm;
- Weight without cryoagent (nitrogen) 30 kg;
- Volume of the cryoagent tank22 L;
- Optical window: diameter 90 mm, material is quartz;
- Electrical connector SNC13-102—3 pieces;
- Pumping flange—KF16.

The main advantages of the system:

- Large-volume nitrogen optical no-spill Dewar;
- Ensures different temperatures of parts of the optical path;
- Minimal temperature is T = 80 K $\pm$ 2 K;
- It is possible to place the cryostat in the primary focus of the BTA at SAO RAS.

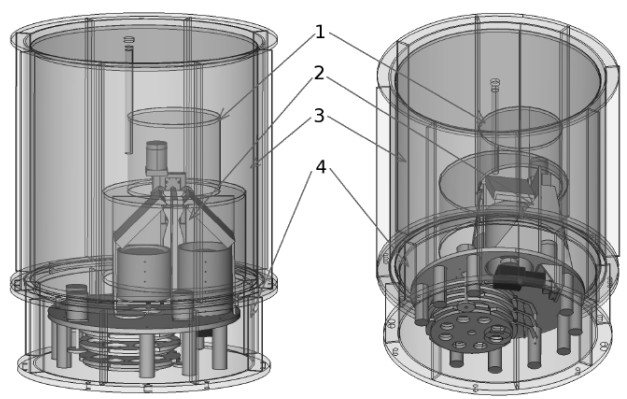 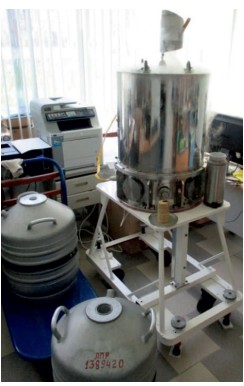

1—nitrogen tank, 2—main frame, 3—upper part of the cryostat body, 4—lower part of the cryostat body

(**a**)

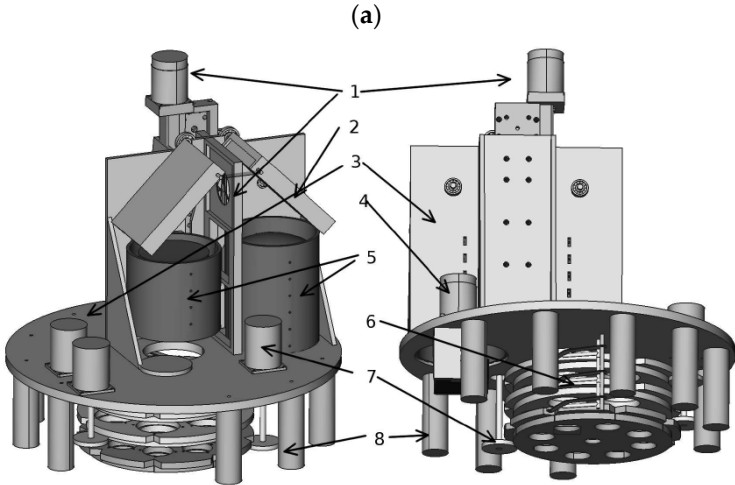

1—movement with a block of slits and an iris mask, 2—diagonal mirrors, 3—the frame of the device, 4—focus movement, 5—collimator and camera, 6—turret block, 7—turret drives, 8—thermal insulation supports

(**b**)

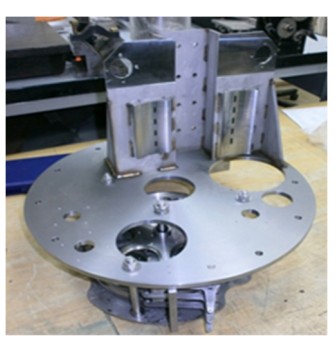 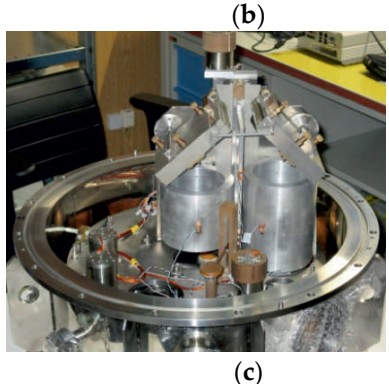 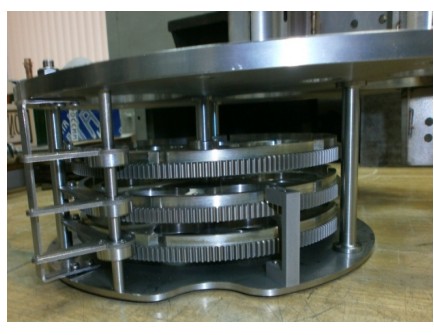

(**c**)

**Figure 4.** Nitrogen optical cryostat: (**a**) 3D model and photograph of the general view of the cryostat; (**b**) general view of the cryostated IR spectrometer; (**c**) photographs of the elements of a cooled IR spectrometer.

### 2.1.5. Systems with Remote Cooling

According to the authors, the most interesting design of a cryostat with a unique set of thermal and optical interfaces was created in SAO RAS and IAP RAS for a two-frequency array receiver with nitrogen cooling and a head with a CCD matrix carried outside of the filler nitrogen cryostat (remote cooling) [29]. The task was to minimize the shading of the

cryogenic system by the elements of the radiation beam in the two-mirror system of the telescope. The solution was to remove the Dewar with nitrogen from the beam zone. At the same time, the head with the matrix had a ratio of the effective area with the matrix to the total area of the head and the cryochannel—cooling wire about 90%. An additional difficulty consisted in sealing the window on an extremely small landing flange. A photo of the system is shown in Figure 5.

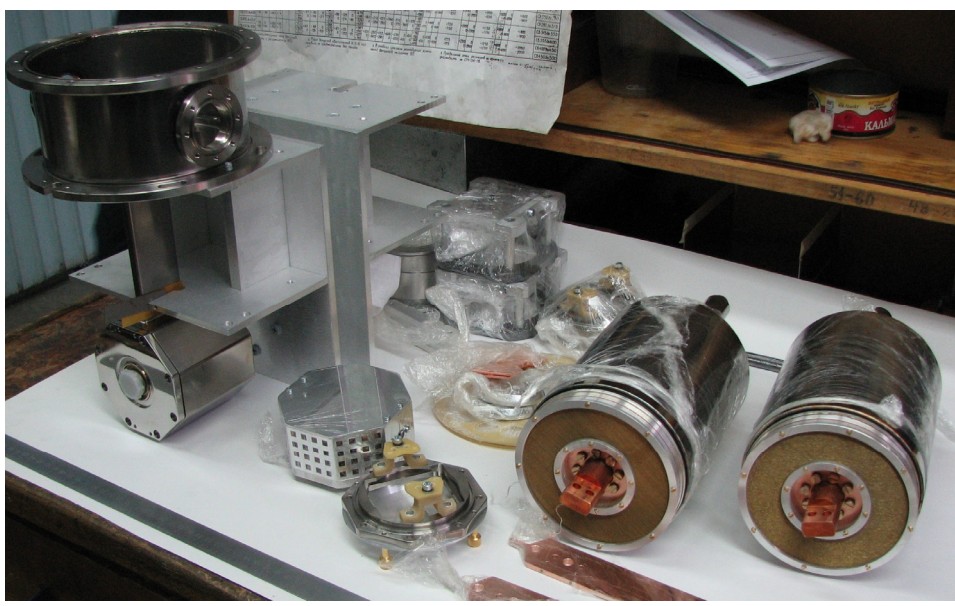

**Figure 5.** The photo of the system with remote cooling.

*2.2. Cooling Systems for Radioreceivers of RATAN-600*

The RATAN-600 radio telescope from the first years of operation on the initiative of D.V. Korolkov [30] implied the use of radio receivers with cryogenically cooled components and the cryogenic department existed as part of the observatory until recently. The ideology of cryogenic refrigerators was chosen as the basis of cryogenic cooling systems, and the cryogenic department was engaged in the operation, maintenance, and repair of refrigerators. Basically, these were hydrogen-level refrigerators manufactured by JSC Cryogenic Equipment (formerly Sibkriotechnika) [31], since the semiconductor receivers, which were cooled mainly, did not require deeper cooling, as compared with the superconducting ones. The vast majority of the SAO RAS receivers are presented in detail in the extensive bibliography and will not be considered here. The three cryogenic radiometers, developed on the initiative with the participation or by order of SAO RAS, which did not work for a single day as part of the RATAN-600 telescope, but largely determined the development of the culture of cryogenic receivers for astronomy, will be presented below.

2.2.1. The 20 K Cryogenic Radiometer of 4 mm Band

At the end of the 70 s, on the initiative of D.V. Korolkov, with the participation of the collaboration of SAO RAS, IAP RAS, and Saturn Research Institute (Kiev), a cryogenic radiometer of the 4 mm wavelength range with a cooled Schottky diode mixer and an intermediate-frequency parametric amplifier was fabricated [26]. The cryostat was made of thick (10–12 mm) stainless steel and was cooled by a Gifford–McMahon refrigerator at a hydrogen temperature level (~12 K); see Figure 6. It demonstrated excellent performance at that time, but the untimely death of D.V. Korolkov prevented it from being installed on RATAN-600. At the same time, the demand for such an instrument was high, and after small modifications, first as a 4 mm and then a 3 mm device, it was used for radioastronomical observations on the RT-22 KrAO radio telescope [25,26]. The complexity and necessity of periodic tuning of the parametric amplifier and the progress in the development of HEMT

(high electron mobility transistor) [32] amplifier technologies soon led to the replacement of PA with HEMT [33], and for a long time determined the architecture of the radio receiver for astronomy up to short-mm or sub-THz waves. At this stage, the world leader in the development of cryogenic receivers for astronomy was Virginia Diodes. In Europe, such receivers were successfully developed by Antti Raisanen [34,35]. This ideology and the components of these SAO receivers were actively, and for a long time, used for other telescopes and applications before the era of superconducting receivers [36] in the sub-THz range began to dominate.

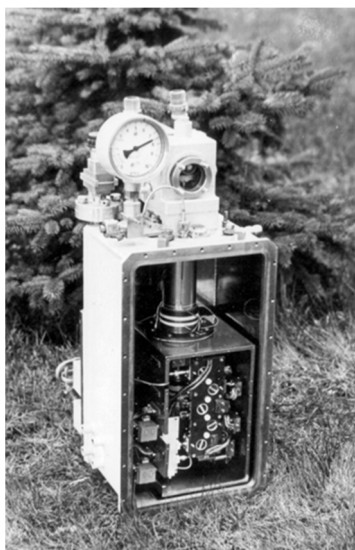

**Figure 6.** Photo of the 20 K cryogenic radiometer of 4 mm band.

The first cryostat (Figure 6) was designed as a vacuum cryostat in the form of a parallelepiped with a volume of about 20 L with thick walls (10 mm) made of stainless steel. A nickel-plated copper radiation shield is visible inside. The shield cover and the cover of the cryostat have been removed. Inside the shield, there is a cooled DBS mixer and a parametric intermediate frequency amplifier. Quasi-optical vacuum windows with antireflective quarter-wave coating are directed opposite to the reader, and their edges are visible at the left edge of the cryostat. The Gifford–McMahon cooler head is visible from above, and the shield and receiver elements are attached to it.

### 2.2.2. The Last Cryogenic Receiver of RATAN-600

The choice of the receiver's physical temperature is a complex multifactorial process. It would seem that the goal in astronomical receivers is the same: to achieve extremely high sensitivity or extremely low noise of receiver. However, in reality, the choice of temperature and cooling system is determined by a compromise between:

- Available cryogenic systems and their maximum cooling levels and cooling capacity, as well as being ready to pay big money for unique systems;
- The development of semiconductor and superconducting technologies of detectors and amplifiers of the corresponding frequency range of observations within a total spectrum from RF to UV waves for SAO RAS;
- Atmospheric limitations of the location of the observatory also determining the effectiveness of the telescope in a particular frequency range.

The first two factors determined the course of the development of cryogenic systems at SAO for many years.

The combination of the second and third factors led about 10 years ago to the liquidation of the cryogenic department in SAO [21]. The fact is that the noise temperatures of uncooled receivers up to the MM range have become very low: units and tens of Kelvin

depending on the frequency. Of course, cryogenic receivers are still much better than uncooled ones and practically reach the quantum limit of sensitivity. But the sensitivity of the receiver [37] is determined not by the noise of the receiver, but (1) by the noise of the system, including the noise of the antenna and the atmosphere above it:

$$\delta T = k \times T_{sys} \times (\delta f \times \tau)^{-1/2} \tag{1}$$

where $\delta T$ is the fluctuation limit of sensitivity; $T_{sys} = T_r + T_a$ is the noise temperature of the system determined as the sum of the receiver noise and the antenna noise, including the atmospheric noise; $k$ is the Boltzmann constant; $\delta f$ is the band; and $\tau$ is the time integration constant.

In fact, with an extremely low receiver noise, the atmosphere can already make a dominant contribution, and the effect of cooling on sensitivity drops sharply. Moreover, the presence of vacuum interfaces in cryogenic receivers, i.e., windows that contribute their losses and noise, negates the feasibility of cryogenics on RATAN-600, located at a low altitude above the sea level (980 m) and having a significant contribution of the atmosphere at 3 mm waves. This fact was indeed the reason for the closure of the cryogenic department and the lack of demand for the last of the mm wave band radiometers developed for SAO RAS, presented below in Figure 7. The last RATAN cryostat, shown in Figure 7, is made of thin steel (1.5 mm) in the original strong and shape-stable topology with approximately the same volume as the first, while containing four receiver channels inside for two frequency ranges and two polarizations. The shape of the cryostat is oval in the cross section, with shortened narrow edges. The cold head of the CryoTiger refrigerator is shown at the top. The horns that feed the RATAN-600 antenna are located on the left, and the blue and orange boxes on the right are the containers for the uncooled components of the receivers.

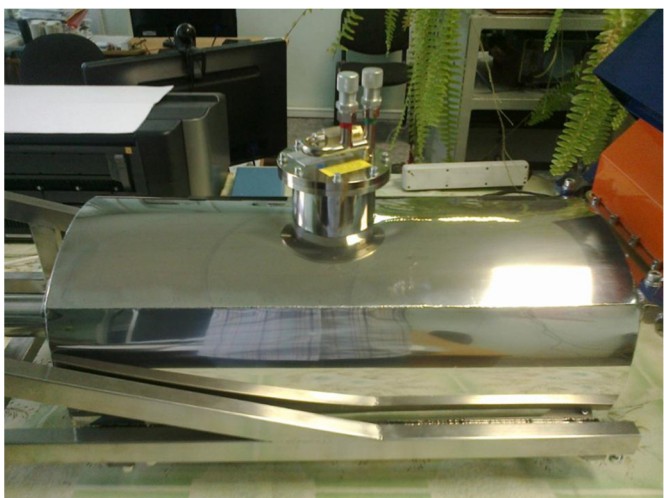

**Figure 7.** The last cryogenic receiver of RATAN-600 at SAO RAS.

Device characteristics:
- Frequency range 18.5–21.5 GHz (center frequency 20 GHz) and 27.5–32.5 GHz;
- The possibility of receiving two mutually perpendicular polarizations of electromagnetic radiation;
- Working temperature is 80 K;
- Construction scheme—modulating with a waveguide ferrite switch and a cooled matched load;
- Fluctuation sensitivity is not worse than 5 mK per 1 s of the time constant of the output filter.

Consideration of the sub-THz range for the RATAN-600 has limited prospects not only because of the small height and large absorption, but also because of the quality of the antenna. It meets the needs of observations of the cm and long mm waves.

However, following the logic of analyzing the ratio of atmospheric noise and receivers, we can confidently conclude that it is advisable to cool the sub-THz radiometers even for low altitudes. In more promising and high-altitude areas, cryogenics will remain a key tool of astronomical observatories for many years to come. The receiver presented here operates in the centimeter range, rather than in the mm or sub-THz one. And the atmosphere affects sensitivity much less than at a wavelength of 3 mm, where it is still rational to use cryogenic receivers.

### 2.2.3. Is Cryogenics Required for Optical Receivers?

The sad conclusion of the previous paragraph that cryogenics is no longer needed for cooling radio receivers for the RATAN-600 up to the mm range raises the question: is cryogenics needed for optical receivers on the BTA? For many years, electric vacuum photomultiplier tubes (PMTs) on optical telescopes successfully solved the problem of astronomical observations in the optical range. Here, the radiation quantum is energetic enough to be reliably detected. The PMT era ended with the introduction of CCD matrices. However, now there are [38] solid-state multipliers that solve a similar problem, not for a single pixel, but for matrices. The signal level after the single-photon receiver and solid-state multiplier would seem to be sufficient even without cooling. However, this turned out to be not entirely true. Cooling, although not cryogenic, is still necessary to minimize dark current. Minus 20 C is still required. And this is not difficult to provide with a simple Peltier cooling machine.

### 3. Results and Discussions

#### 3.1. Results of the Development of Cooling of Optical Receivers for an Optical Telescope and Radio Receivers for a Radio Telescope

Results have been achieved in the development of cryogenic systems for cooling photodetectors for the BTA optical telescope and radio receivers for the RATAN-600 radiotelescope. A total of over 20 [22] cryosystems of various temperature levels from nitrogen to helium were created and successfully used for astronomical observations on the both telescopes in the radio range up to 3 mm and in optics, including IR. There are lots of new astrophysical results obtained due to the use of the developed receivers:

- Direct evidence of the fossil origin of large-scale magnetic fields of chemically peculiar stars obtained [39];
- The discovery of new LBVs in the Local Volume galaxy NGC 1156 [40];
- Planet TOI1408.01: a grazing transit and probably a highly eccentric orbit [41];
- Collected RATAN-600 multifrequency data for the BL Lacertae objects [42];
- The RATAN-600 telescope helps to understand the origin of cosmic neutrinos [43];
- Identified Quasi-periodic Pulsations in a Solar Microflare [44].

#### 3.2. Cooling Systems of the Sub-THz Range for Radioreceivers as a Part of the BTA Telescope

Within the framework of Project 23-62-10013 of the Russian Science Foundation "Development of Russian sub-terahertz observatory prototype as part of an optical telescope", the authors of the article set themselves the task of expanding the capabilities of the optical Big Telescope Alt-Azimuth (BTA) of SAO RAS in the sub-terahertz range. In this article, we will not present the entire project in detail, but the issue of creating one of the key nodes, specifically, a specialized sub-Kelvin cryogenic system and a set of its interfaces with elements of the receiving complex and the telescope itself will be considered in detail. The key motivation of this project and this article was the intention to place a sub-THz receiver at the focus of the BTA optical telescope. Highly sensitive bolometric receivers of sub-THz waves require deep sub-Kelvin cooling.

The first experience of placing a sub-THz receiver on a cooled Shottky barrier diode mixer that was developed at IAP RAS for BTA dates back to 1991. The experience was not successful, but led to the intensification of cooperation between SAO RAS and IAP RAS, which gave rise to a series of successful projects of cooled photodetectors presented in Section 3.1.

We returned to this idea [45]. Two samples of a sorption cryostat with an operating temperature of 300 mK were manufactured (photo in Figure 8).

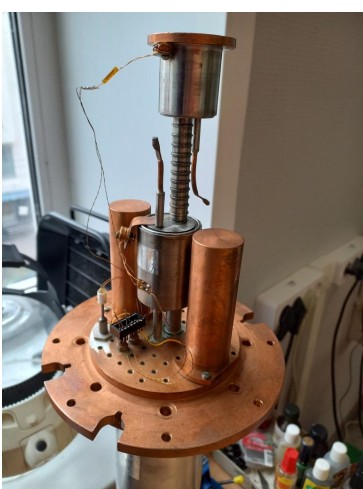

**Figure 8.** Sorption cryostat with an operating temperature of 300 mK.

The main characteristics:

- Operating temperature: 300 mK ± 0.1 K;
- Heat load: 1 mW;
- Vacuum level: $10^{-4}$ mbar;
- Two flanges for the installation of optical windows with a diameter of 25 mm;
- Flange KF D25 for pumping;
- Input of electrical and RF connectors;
- The size of the working cavity: diameter 185 mm, height—70 mm, dimensions—not more than 1600 mm;
- Diameter: 700 mm.

The main advantages:
Advantages of sorption pumping:

- Compact placement inside the cryostat;
- Possibility of automation;
- Low cost compared to mechanical pumping means.

Advantages:

- A two-stage sorption pumping system has been developed;
- The gas heat key is integrated with the sorption pump into a single unit;
- The technology of manufacturing sorption pumps makes it possible to obtain a high degree of pumping compared to Western analogues;
- Two models were developed: with filler and using a closed-loop cryogenic system.

The second sample of a similar cryosorption system contained a precooling system based on a 4 K closed-cycle refrigerator, and not in the form of a filler helium cryostat. However, it did not go beyond a series of laboratory tests. The problem was the lack of progress in developing reliable, reproducible, and capable of folding into matrices of identical pixels high sensitive detectors. At the moment, the problem has been solved and a series of both single detectors, such as in [45], and matrices, which are not difficult to modify for the BTA telescope irradiation system, has been created. Over the next two years

of the project, the detectors will be finalized, the receiving path will be built and integrated into the BTA optical system, the cryogenic system and the necessary interfaces will be created. In particular, one of the 0.3 K variants of the BTA receiver cooler is built on the basis of the refrigerator shown in 8, but with the replacement of the Dewar with a 4 K refrigerator head.

### 3.3. Sorption Cryostat

One of the first mentions of the creation of a 0.3 K cryogenic system for installing of a sub-THz receiver on BTA is given in publication [45], and the system was at the design stage. He3-He4 cryogenic sorption and dilution systems give unique possibilities to cool down astronomical receivers to the temperature level below 1 K [46–50]. In this article, a cryogenic system was proposed that provides the operating temperature of the receiving antenna arrays T~0.3 K. It is a closed-cycle system, i.e., it does not require the use of liquid gases (nitrogen and helium), based on a refrigerator on pulse pipes and two sorption pumps on $^3$He and $^4$He. The system includes a refrigerator based on pulse tubes. Despite the lack of funding for this project at that time, work on the creation of such a cryogenic system was continued, and the next stage of development was already presented in publication [51].

The project proposes the development and installation of a cooled receiver based on SINIS [52] (superconductor–insulator–normal metal–insulator–superconductor) detectors at the BTA site of SAO RAS.

Preliminary assessments of astroclimatic studies at the BTA site showed that the most successful range for test studies of the receiving system is 3 mm, since observations can be carried out year–round in this window of transparency. Also, in the future, transparency windows of 1.3 mm and possibly 0.8 mm will be considered (however, in this case, favorable conditions account for only 2% of the time a year in winter, or observations should be directed to a powerful radiation source). The optimal operating temperature of the designed SINIS detectors is 300 mK or lower (to improve sensitivity), which requires a complex cryogenic system. It should be noted that over the past fifteen years, cryosorption cryostats and dissolution cryostats have become a common commercial product. Oxford Instruments, Bluforce, and others offer [53] convenient commercial products to achieve these temperatures with reasonable cooling capacity of units and tenths of a microwatt. However, these are large and extremely inconvenient machines for mounting in the focus of the telescope, and in fact it is only a tool for laboratory experiments. For installation on a telescope, it is necessary to develop a specialized cryostat and, most importantly, a set of highly efficient interfaces, in particular mechanical, vacuum, cryogenic, optical, electrical ones, etc., that are capable of providing effective cooling of the detector cooling object, as well as the supply/removal of signals to it without loss and with maximum thermal insulation from the environment. At the moment, three variants of cryogenic systems are being considered, the descriptions of which are given in this section. The first one is based on the 2008 groundwork described above (and presented in Figure 9).

### 3.4. Cryogenic System for Sub-THz Detectors for the BTA Based on a Dilution Microcryostat–Deep Stick

The dilution microcryostat–deep stick [54] (Figure 10) was developed by V.S. Edelman at the Kapitsa Institute of Physical Problems of the Russian Academy of Sciences. The minimum achievable operating temperature in such a cryostat can reach 50 mK (without a background load in the absence of an optical window) and nanowatt capacity. In addition to the achievability of such low temperatures, another advantage of this device is the absence of pulse tubes (which contribute to the total noise of the receiving system) because pumping of $^3$He and $^4$He is carried out using built-in sorption pumps. But there is also a certain drawback of such a cryogenic system, specifically that the duty cycle is no more than 8 h. This generally fits into the ideology of using time unoccupied by night optical observations for sub-THz observations on an optical telescope. At night, when observations

are made in optics, $^3$He will be recondensed. The dilution cryostat is controlled by sorption heaters, sorption gas heat key heaters and evaporator heater.

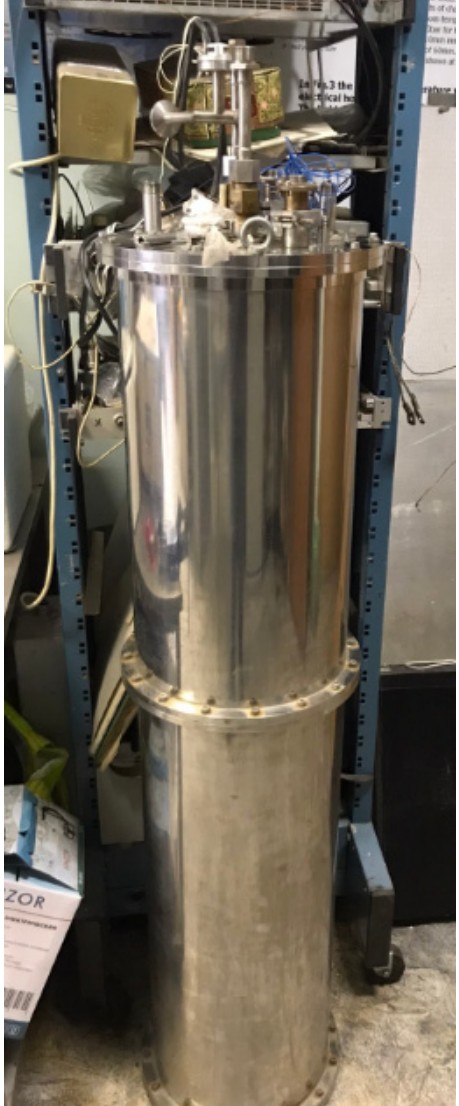
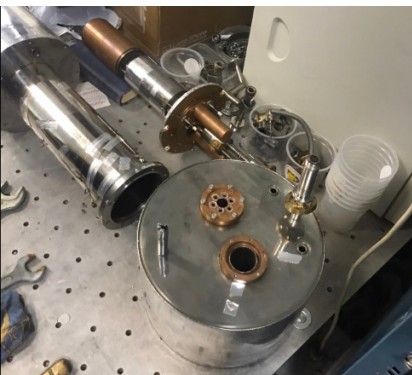
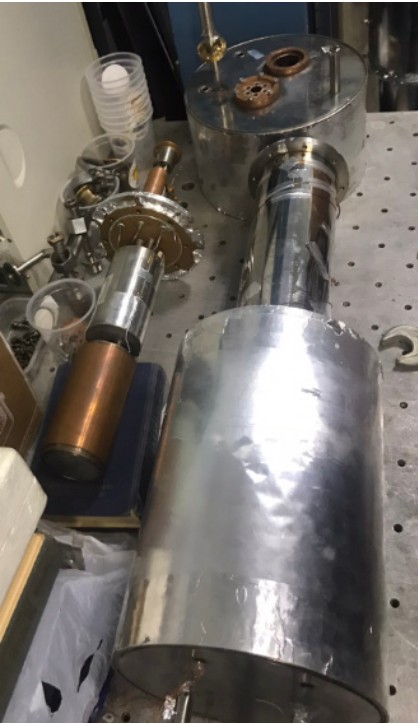

**Figure 9.** The 0.3 K sorption cryostat.

Such equipment is promising for conducting short-term observations, but requires high sensitivity of the receiving system (due to the absence of external vibrations from pulsating pipes and low operating temperature).

### 3.5. A Continuously Operating Dilution Microcryostat

To ensure long-term temperature maintenance of the operating temperature of the detector, a modification of the cryogenic system presented in the previous section was carried out [54]. The principle of operation is based on the technology of sorption pumping $^3$He and $^4$He and condensation pumping of the mixture $^3$He–$^4$He in the dissolution cycle. The continuous maintenance of a low temperature of the order of 0.1 K during the regeneration of sorbents is carried out due to the large heat capacity of the block filled with holmium plates, in which the condensation of circulating helium occurs.

The structurally presented cryostat is in many ways similar to the designs given in the works [54,55]. The cryostat is made in the form of an insert (deep stick) into an industrial 35 L transport liquid helium Dewar. The cold sample holder is located at the top of the

device, which facilitates access to it. A total of 35 L of liquid helium is enough to precool the device from nitrogen temperature and for 6–7 days of operation. During this time, a low temperature of the sample holder can be maintained with accuracy of the order of a millikelvin.

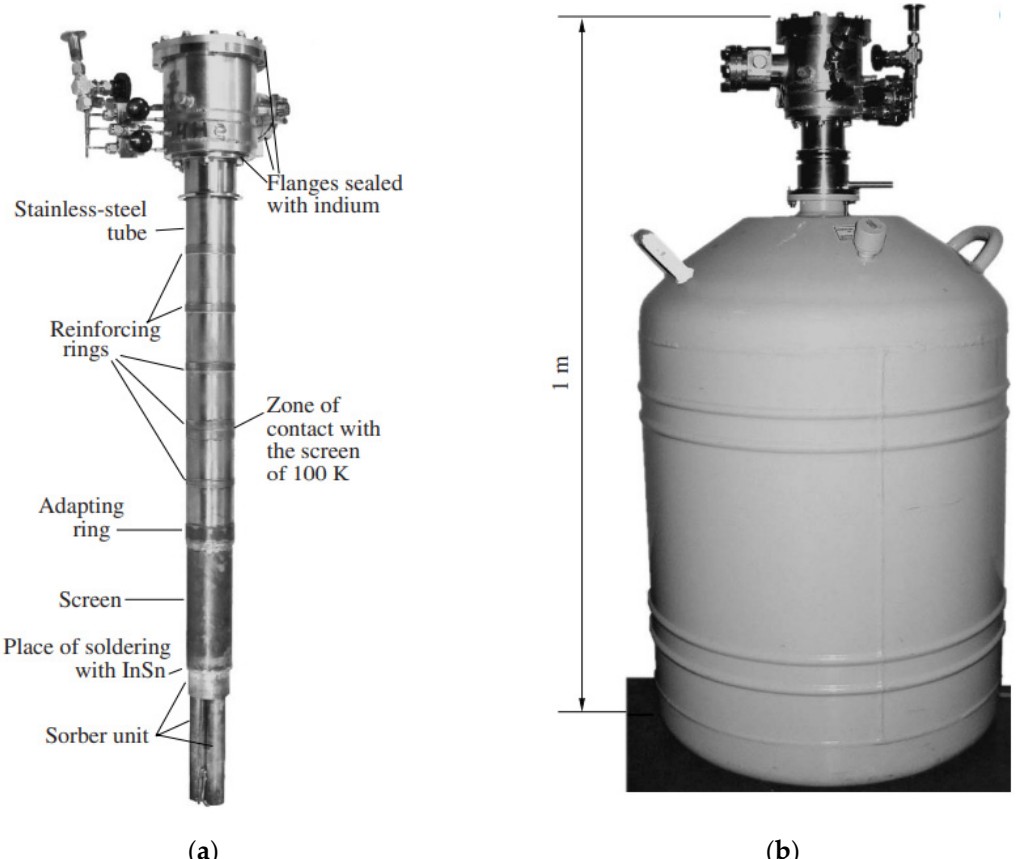

(**a**)　　　　　　　　　　　　　　　　　(**b**)

**Figure 10.** (**a**) Appearance of the cryostat–deep stick mounted in (**b**) a portable helium cryostat with a volume of 35 L.

A simplified schematic representation of the structure is shown in Figure 11. The microcryostat contains the following functional units located in a common vacuum volume:

- The $^4$He unit that includes a 1 K chamber and a $^3$He condensation volume, which constitute a common unit, and a sorption pump that evacuates helium vapors. The $^4$He sorber (as well the $^3$He sorber) is above the $^4$He bath and both $^3$He baths. Heat is withdrawn from the sorber to liquid helium in the portable Dewar flask through copper heat conductors. The sorber on which a heater is wound is manufactured from stainless steel and placed in a sealed copper container. The evacuation channel consists of a stainless steel tube into which a copper tube about 5 mm long is soldered. The latter passes through the container cover and is soldered to it. In this tube, $^4$He condenses and trickles down to the 1 K chamber. A thermal valve, during whose heating/cooling the heat exchanging gas ($^3$He) is desorbed/sorbed, serves for controlling the heat exchange between the sorber and the container walls. This is necessary when changing from the $^4$He desorption regime to its evacuation.
- The upper $^3$He unit that contains the $^3$He sorber and the condenser of $^3$He vapors, which is soldered into it and serves for the lower $^3$He bath. The $^3$He bath is cooled by the evacuation of vapors of liquid $^3$He by its own sorption pump, which is analogous to the $^3$He pump. When the pump is regenerated, the bath is filled with $^3$He, which is liquefied when being in contact with the 1 K chamber.

- The lower $^3$He unit that contains the second $^3$He bath, which is under the first bath and is connected to the condenser of $^3$He vapors of the upper circuit with a stainless steel tube. The both circuits are sealed with respect to each other. If the temperature of the upper bath is lower than that of the lower bath, a good thermal contact establishes between them; otherwise, the thermal interaction is substantially weakened.
- The dilution unit that contains a dilution bath (mixer), a heat exchanger, a still, and a condenser of vapors of a $^3$He–$^4$He mixture. The condenser is filled with plates of holmium that has the high specific heat at T < 1 K. For a He mass of ~50 g, its specific heat at T = 0.4 K is ~2 J/K. This allowed us not to use a large amount of $^3$He for maintaining the working conditions during the regeneration of $^3$He in the upper circuit, and to restrict ourselves to a rather small amount of $^3$He, only 0.015 mol in the lower circuit for establishing thermal coupling. The dilution circuit is placed above the sorption pumps, and the condenser included in it is connected to the lower $^3$He bath via a copper heat conductor. The temperature difference between the upper $^3$He bath and the condenser at T = 0.4 K is approximately 0.02 K/100 µW.

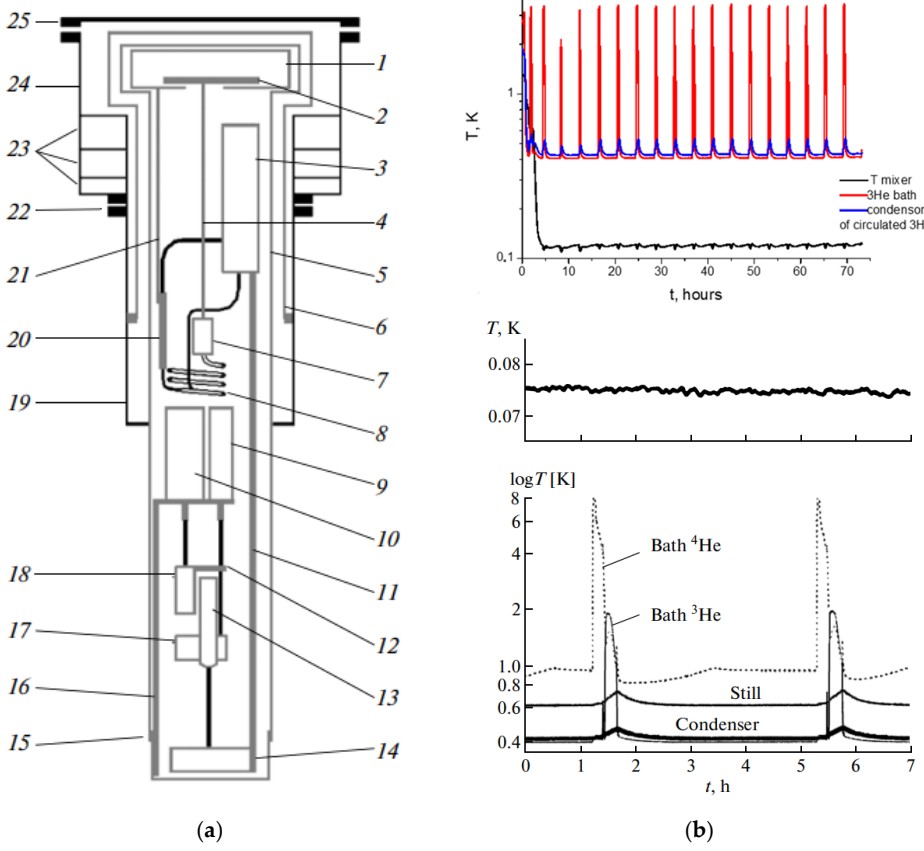

(a) (b)

**Figure 11.** A continuously operating dilution microcryostat (**54**). (**a**) Simplified diagram of the microcryostat that shows the arrangement of the main units: (1, 5, 6) 0.6, 4.2, and 100 K shields, respectively; (2) platform at 0.06–0.1 K; (3) holmium-filled condenser of the mixture vapors; (4, 11, 16, 21) 0.1, 0.4, 4.2, and 0.6 K copper heat conductors, respectively; (7) mixer; (8) tubular heat exchanger; (9) $^3$He sorber; (10) $^4$He sorber; (12) condenser of $^3$He arriving at the upper bath; (13) $^3$He condenser of the lower circuit; (14) lower $^3$He bath; (15) vacuum-tight soldered spots of shield 5; (17) upper $^3$He bath; (18) $^4$He bath (1 K chamber); (19) outer stainless steel tube with a 56 mm diameter; (20) still; (22) flange of the outer tube; (23) vessels for collecting the working gases that are stored in them at a pressure of 25–50 atm at room temperature of the instrument; (24) housing; and (25) detachable cap. Stainless steel tubes and parts are black-colored; copper parts are gray-colored. (**b**) Time dependences of the temperatures of the mixer (upper curve) and the $^4$He bath, upper $^3$He bath, still, and condenser of the mixture vapors (lower curves).

### 3.6. Dilution Microcryostat with Cooling by a Refrigerator with a Pulse Tube

More promising for autonomous operation on the BTA telescope is the dilution microcryostat designed for scientific research at temperatures up to 0.1 K and below. To ensure its operation, a refrigerator with a pulse tube is used. It is possible to maintain the temperature at 0.1 K for 5 h with the refrigerator not working. The device is used to study low-temperature radiation receivers. The photo of dilution microcryostat with cooling by a refrigerator with a pulse tube is presented in Figure 12.

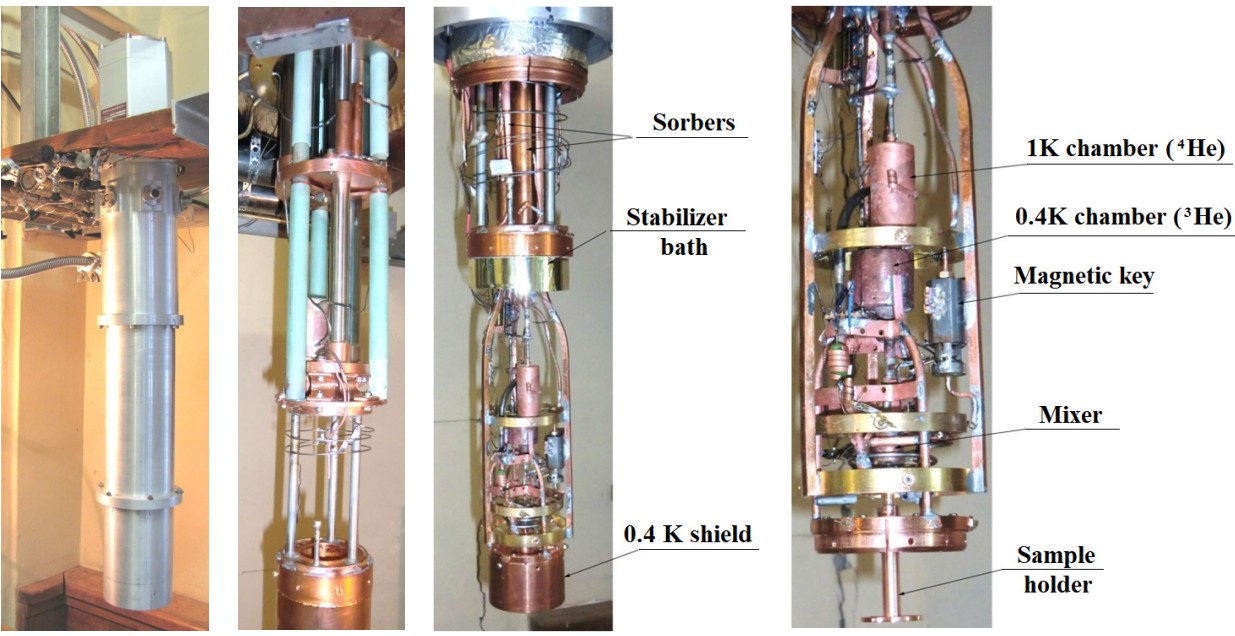

**Figure 12.** The photo of dilution microcryostat with cooling by a refrigerator with a pulse tube.

The main characteristics of the system:

- Minimum fixed temperature of the sample holder: 0.07 K.
- Performance at 0.1 K: about 1–3 μW.
- Cooling time to 0.1 K (from the start of pumping $^3$He): 30–40 min.
- Time of regeneration of sorbers, condensation, and cooling $^3$He up to 1 K: about 1.5 h.
- Time of temperature maintenance below 0.1 K: 4–6 h.
- Time to maintain working conditions when PT is turned off: up to 8–9 h.
- The time to restore the initial conditions after a new start PT: approximately 1 h.
- Cooling time from room temperature to 0.1 K: 10–12 h.
- Quantity of gases used: 4.5 n.l. $^4$He; 3 n.l $^3$He; 1 n.l of mixture 30% $^3$Ha + 70% $^4$He.
- Energy and water consumption during PT operation according to the manufacture's passport: 6–8 kWh and 8–10 L/min.

In essence, the latter option, called the "liquefaction circuit", is the most promising, because it combines the advantages of a refrigerated circuit and requires filling of a cryoagent. But at the same time, for a long time (up to 8–9 h), it is able to maintain the operating temperature with the compressor and cooler turned off, i.e., absolutely without vibrations and noises and temperature fluctuations caused by the frequency of operation of the cooler with a frequency of about 1 s. Refrigeration systems without a liquefaction circuit, operating continuously generate noise and temperature fluctuations that dramatically affect the characteristics of the receiver and require special measures.

The main goals of this work have been achieved. At the same time, fixing certain specific temperatures is not our scientific achievement. The temperatures are just normal and usual for cryogenics. Today, cooling records are below nanokelvin. Our achievements lie in the fact that a sufficient temperature has been reached to achieve a certain level of

sensitivity of receivers or a state of reliable superconductivity of superconducting detectors, that is, to ensure their operability. For cooled photonics, this means the disappearance of dark current and improved image quality. We would like to emphasize that an important feature of the cryogenic systems being created is their high level of functionality and availability. In particular, reaching a given temperature cooling regime takes about one hour for cryosystems to the hydrogen level and several (2–4) hours for helium and subhelium machines. This ensures ease of use of telescopes and does not take up a significant part of the time of astronomical observations. Another important result is the successful spatial, thermal, and optical matching of the developed cryogenic systems with built-in receivers and optical-mechanical structures of telescopes.

## 4. Conclusions

This paper provides a brief overview of the development of cryogenic systems operated at different temperature levels and the cooling principles for cooling radio and photo receivers of various frequency ranges for astronomical research using the main instruments of the Special Astrophysical Observatory of the Russian Academy of Sciences, specifically the BTA optical telescope and the RATAN-600 radio telescope.

The main results of the presented work are as follows:

-   A successful line of cryogenic systems actually working at various temperature levels has been developed actively used over the years, and is now used for cooling photodetectors in the BTA telescope and radio receivers for the RATAN radio telescope;
-   This work presents a new unique project of a cryosystem for the ultradeep cooling of a radio receiver for the BTA optical telescope;
-   Using these instruments, new unique astronomical results were obtained and presented in a wide list of highly rated publications reviewed in [39–44,56];
-   Dozens of highly efficient and reliable cryogenic systems have been developed that cool highly sensitive radio and photodetectors of the two telescopes at the Special Astrophysical Observatory of the Russian Academy of Sciences. Using these instruments, new unique astronomical results were obtained, presented in a wide list of highly rated publications reviewed in [39–44,56].

This is the first presentation of the new project connected with the development of a sub-THz radio receiver for the optical BTA telescope. Several variants of cryogenic systems of the sub-Kelvin level for operation of the sub-THz receiver of the BTA optical telescope at SAO RAS have been proposed for the new project.

The proposed approach will make it possible to fill part of the gap in the spectrum of astronomical research between the RATAN-600 observation windows in the radio range and the operating range of the receivers of the BTA optical telescope, of course, taking into account the limited transmission of THz waves shorter than 1 mm at the location of SAO RAS. There is a real astronomical science for such combination of optical telescope and radio receiver [56].

**Author Contributions:** Conceptualization, Y.B. and G.V.; methodology, S.M., N.T. and V.E.; software, O.B. and I.L.; formal analysis, V.V.; investigation, A.G., A.V., M.M. (Maria Markina), A.E., E.E., G.M. and A.K.; design and hardware, M.M. (Maria Mansfeld), E.P. and A.C.; writing—original draft preparation, A.G. and V.V.; writing—review and editing, G.V. and V.V.; supervision, Y.B.; project administration, Y.B. and V.V.; funding acquisition, Y.B. All authors have read and agreed to the published version of the manuscript.

**Funding:** The development of cryogenic systems for cooling of the sub-THz receiving device on the BTA telescope of SAO RAS is supported by the grant of the Russian Science Foundation # 23-62-10013 "Development of Russian subterahertz observatory prototype as part of an optical telescope".

**Institutional Review Board Statement:** Not applicable.

**Informed Consent Statement:** Not applicable.

**Data Availability Statement:** Data are contained within the article.

**Acknowledgments:** The authors are grateful to the employees of SAO RAS, IRE RAS, and IAP RAS who are currently working and have left us for the persistent development of cooled receivers for radio astronomy, as well as colleagues from the Cryogenic Technology Research Center (Omsk) and the Saturn Research Institute (Kiev) who actively contributed to these works. Cryogenic equipment was developed and manufactured on the basis of Large-scale research facilities «Center of Microwave Research of Materials and Substances» (CKP-7, UNU # 3589084).

**Conflicts of Interest:** The authors declare no conflict of interest.

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
