# Peer review of "Cryogenic Systems for Astronomical Research in the Special Astrophysical Observatory of the Russian Academy of Sciences"

_photonics, doi:10.3390/photonics10111263_

Round 1

Reviewer 1 Report

Comments and Suggestions for Authors

This paper is a summary of current work in the SAO on cryogenic systems for radio astronomy, leading to mm-wave receivers located on the BTA optical-IR telescope. In order to be useful to the reader, however, significant changes are required.

The paper should be restructured to ensure that the reader understands the logic of the paper. What is the underlying scientific interest? And how does this drive instrumentation choices? In the abstract, the authors mention ALMA and EHT. But it's very unlikely that the instrumentation under development will do similar science. Is the instrumentation continuum-sensitive or for spectral lines. The extensive work on sub-K cryogenics suggests continuum, since spectral-line receivers are generally only cooled to about 4K. But it's not clear from the paper; it seems to be implied in l.79

Section 3.4 talks about bolometric receivers, and this seems to be the first point where it's made clear that bolometric receivers are the focus of the paper. It needs to be explained what the scientific reason is for implementing bolometric receivers. At a marginal site, such as BTA, it's much harder to get good results from bolometric receivers than from heterodyne receivers - so what is the driver for using bolometers?

It would also be helpful to explain what science is likely to be enabled by mounting a mm-wave bolometer on a 6-m mirror.

The paper should make it clear from the start that the aim is to build mm-wave bolometer receivers for the BTA Optical/IR telescope.

The authors should engage with the non-Russian literature. A great deal of work has been done in this field in the USA, Europe and Japan. The Russian literature citations are welcome, especially for eg the astroclimate work - but other literature needs to be consulted too.
For example, Optical-IR telescopes have been used for sub-mm work for a long time, see eg White et al 1981, MNRAS 197, 745.

There are a number of points that are incorrect in detail, which I've noted below. Also some points where specific explanations should be given.

There is also quite a lot of detail given that doesn't seem relevant. I believe the paper would benefit from being shortened quite significantly. Each part of it should be checked to see whether it is truly relevant to the actual point of the paper, which is the development of mm-wave receivers for BTA.

Specific points:

l.61 The list of places that can handle high-frequency sub-mm waves (around 800 GHz) should include Mauna Kea, Hawaii, and the Antarctic Plateau.

l.62 If you're going to call it the subTHz range, your windows should be given in frequency as well as wavelength

l.66 'remote from large water spaces' is incorrect. Mauna Kea is surrounded by water, and the Andes mountains are not far from the Pacific Ocean.

l.67 it would probably be helpful to give more detail of the constraints imposed by the astroclimate - what windows can possibly be used at BTA, and what windows are not possible?

l.87 I don't think this is correct. In my experience, cooling with LN2 and LHe is a routine part of working in these frequencies. It is only recently that closed-cycle cooling has started to replace liquid cryogens.

l.99 I'm not familiar with LH2 cryostats, but I would suspect that they are not used because SIS junctions need to go below about 7K to become superconducting, while bolometers need to go much colder. 20K is only interesting for e.g. Schottky diodes or as a cold shield.

l.105 A reference would be helpful

Section 3.1; given that the point of the paper is the development of mm-wave receivers for BTA, why is the cooling of CCD arrays at BTA relevant? This section should be removed, or shortened with a clear explanation of its relevance.

ll. 337-346: what device is this referring to? The previous paragraph has been talking about a 3mm/4mm receiver, but this seems to be talking about a 10-13mm receiver. Please clarify or remove.

ll. 478-9: large-scale refrigeration systems e.g. dilution refrigerators have been deployed on telescopes, e.g. SCUBA on the Nasmyth focus platform at JCMT.

ll. 502-3: the night is also the best weather for mm-wave and sub-mm. At BTA, daytime observations at 3mm are probably feasible, but 1.3mm and 0.85mm will likely have to be carried out at night, or around sunrise.

Sect. 4. Conclusions. Again, it should be made clear that the focus of the paper is on cryogenic instrumentation at BTA, not at RATAN-600.

Minor points:

l.51 I'm not familiar with RATAN-600, but my reading indicates that its highest frequency is about 10GHz, several cm. It's not a mm-wave telescope.

l.288 the references should be in []

l.321 please define all terms in this equation

l.350 please define a reasonable frequency/wavelength limit for RATAN600

l.566 'holmiumfilled' -> 'holmium filled'

'subTHz' is an unusual phrase. This waveband is usually measured in GHz, microns, or mm. Given that the focus of the paper is around 1mm wavelength, subTHz, while accurate, is a bit misleading. I would suggest 'mm-wave', as a much more easily-understood term.

Comments on the Quality of English Language

The English is generally good. There are a number of typographical errors, e.g. 'stainlesssteel' without a space. The main space for improvement is in conciseness.

Author Response

Pls.,see the attached file

Reviewer 2 Report

Comments and Suggestions for Authors

This paper demonstrates the development of cryogenic systems of different temperature levels and cooling principles for cooling radio and photo receivers of various frequency ranges for astronomical research, using the main instruments of the Special Astrophysical Observatory of the Russian Academy of Sciences. My comments are given below.

1. Reducing temperature is an effective way to improve the signal-to-noise ratio of astronomical detectors. A series of detector cooling techniques and equipment used for both telescopes are given in this article, but the results achieved with these devices are not described.

2. Different refrigeration equipment should be related to the corresponding needs. The requirements of the refrigeration technology and the effect of the application should be briefly given, before showing each refrigeration technology and equipment.

3. The article lists a large number of refrigeration equipment and the indicators achieved, but there is no relevant supporting materials. It is suggested to add.

4. The paper gives a large number of research results on cryogenic refrigeration equipment, based on the detecting requirements of BAT and RATAN-600 telescopes. However, the paper does not describe the detection results of these two telescopes.

Reviewer 3 Report

Comments and Suggestions for Authors

Comments on the Quality of English Language

Moderate editing of English language required.

Round 2

Reviewer 1 Report

Comments and Suggestions for Authors

I thank the authors for their detailed reply, which has given me a better insight into the structure and purpose of their paper. I still find that it needs minor revision. It's hard to tell exactly what changes have been made, since they have not been clearly shown in the revised version. (It would be helpful for the authors to supply a revised version with changes marked up, eg in bold.)

In their reply, the authors note their long-term interest in expanding the EHT to NE Eurasia. This provides a solid context for the paper, and should be stated clearly in the paper. It motivates the rest of the paper, since a small mm-wave telescope in the region (eg the BTA) can be worthwhile as part of a larger project to a larger purpose-built EHT component on a better site. This goes some way to answering the question: 'What is this for?'.

The authors note that this paper is now the first of two, which together provide a comprehensive review of the work they've done. This does explain the large amount of detail that is marginally relevant to the focus of the paper. Under the circumstances, I think it is necessary that the second paper be explicitly noted in the Introduction. State what is in Paper II and how it relates to the current paper. This is standard practice for a series of papers that are greater than the sum of their parts.

I'm happy to see an explicit statement of the planned instrumentation in lines 91-106

Thank you for the extra detail on LH2 cryogenics and cryocoolers - I learned a lot! The point that cooling of eg HEMTs can give significant performance advantages is well made. Please put it in the paper! When talking about the cryogenics that you're using, make sure that you tell the reader what it's needed for. E.g. sub-K cooling for bolometers; few-K cooling for superconducting heterodyne receivers; and higher temperatures for SBDs and HEMTs. This gives the reader a better context to understand what you've done.

l 323-384 give the details of the last receiver. This is now more clearly separated from the previous section, which is welcome. However, l355 talks about the contribution of the atmosphere at 3mm, when the receiver is working at about 10mm. The atmosphere at 10mm is very different to that at 3mm. I guess the point is that a 3mm receiver at RATAN has so much atmospheric noise that there's no point in making it cryogenic? (This only makes sense, of course, if the receiver is a Schottky diode; if it's SIS, you need cryogenics anyway.) At any rate, an argument about the 3mm atmosphere doesn't apply to a 10mm receiver. You need to make a clear distinction between the problems at 3mm and the design and construction of the 10mm receiver.

l 485 Please specify that it is not convenient to place a commercial dilution or sorption system at the focus of the BTA telescope (and state which focus). We agree that it is possible to mount a dilution refrigerator at the Nasmyth focus of a large telescope.

l 510-1: Given the authors' reply to my comment on observing at twilight and night time, would it not make more sense to recondence the 3He during the daytime? You'd have a lot of time between morning and evening twilight, and it can be done during normal working hours. And you'd be ready to use light-cloud night-time conditions.

l 609 'liquefaction circuit'

Comments on the Quality of English Language

There are some minor errors in English - see some of my comments above. In general, it's fairly clear. Avoid use of phrases like 'It should be noted' - these are meaningless.

Author Response

Thank you!

Pls., find attached

Reviewer 2 Report

Comments and Suggestions for Authors

All my concerns are addressed. The paper can be accepted.

Author Response

Thank you very much!

Reviewer 3 Report

Comments and Suggestions for Authors

Comments on the Quality of English Language

I suggest the Authors to have the paper read by a native speaker before submission.

Author Response

Thank you

Round 3

Reviewer 3 Report

Comments and Suggestions for Authors

Dear Author,

Thank you very much for addressing all my comments. The paper is well revised and I think that it can almost be accepted for publication. Anyway, just a little revision about references needs to be addressed. I recommend to number each Reference in order of appearance in the text.

Comments on the Quality of English Language

Minor editing of English language required

Author Response

Thank you!

See attached comments

V
